# Protective Actions of α-Tocopherol on Cell Membrane Lipids of Paraquat-Stressed Human Astrocytes Using Microarray Technology, MALDI-MS and Lipidomic Analysis

**DOI:** 10.3390/antiox11122440

**Published:** 2022-12-10

**Authors:** Laura Sánchez-Sánchez, Roberto Fernández, Maria Dolores Ganfornina, Egoitz Astigarraga, Gabriel Barreda-Gómez

**Affiliations:** 1Research and Development Department, IMG Pharma Biotech S.L, 48160 Derio, Spain; 2Instituto de Biología y Genética Molecular (IBGM), Unidad de Excelencia, Universidad de Valladolid-CSIC, 47003 Valladolid, Spain

**Keywords:** oxidative stress, lipid peroxidation, membrane microarrays, human cell line, astrocytes

## Abstract

Cellular senescence is one of the main contributors to some neurodegenerative disorders. The early detection of senescent cells or their related effects is a key aspect in treating disease progression. In this functional deterioration, oxidative stress and lipid peroxidation play an important role. Endogenous antioxidant compounds, such as α-tocopherol (vitamin E), can mitigate these undesirable effects, particularly lipid peroxidation, by blocking the reaction between free radicals and unsaturated fatty acid. While the antioxidant actions of α-tocopherol have been studied in various systems, monitoring the specific effects on cell membrane lipids at scales compatible with large screenings has not yet been accomplished. Understanding the changes responsible for this protection against one of the consequences of senescence is therefore necessary. Thus, the goal of this study was to determinate the changes in the lipid environment of a Paraquat-treated human astrocytic cell line, as a cellular oxidative stress model, and the specific actions of the antioxidant, α-tocopherol, using cell membrane microarray technology, MALDI-MS and lipidomic analysis. The stress induced by Paraquat exposure significantly decreased cell viability and triggered membrane lipid changes, such as an increase in certain species of ceramides that are lipid mediators of apoptotic pathways. The pre-treatment of cells with α-tocopherol mitigated these effects, enhancing cell viability and modulating the lipid profile in Paraquat-treated astrocytes. These results demonstrate the lipid modulation effects of α-tocopherol against Paraquat-promoted oxidative stress and validate a novel analytical high-throughput method combining cell cultures, microarray technology, MALDI-MS and multivariate analysis to study antioxidant compounds against cellular senescence.

## 1. Introduction

Lipid peroxidation is a physiological chain reaction process produced by free radicals [1,2] through enzymatic and non-enzymatic mechanisms [3]. In enzymatic mechanisms, lipid peroxidation can be produced by lipoxygenases (LOX), cyclooxygenases (COX), cytochrome c [4], or cytochrome P450 [5,6]. Whereas, in non-enzymatic mechanisms, the production is mediated by oxygen-free radicals [7], such as superoxide ($O_{2}^{•-}$) or hydroxyl radicals (${\mathrm{OH}}^{•}$), among others [8]. However, lipid peroxidation is regulated by a diverse array of antioxidant defense systems, such as the glutathione system or catalase and superoxide dismutase enzymes [6,9], as well as non-enzymatic compounds, such as vitamin E, beta-carotene or glutathione [6]. If an imbalance between the oxidant and antioxidant compounds is produced, it turns into a pathological process, which provokes the modification of physical properties of membranes, such as permeability, packing of lipids and proteins [10], and a loss of function [4] that leads to cell damage [7].

The non-enzymatic formation of lipid radicals is a three-phase reaction (initiation, propagation, and termination) which is primarily produced in polyunsaturated fatty acids (PUFAs) [11], due to its relative weak hydrogen-carbon bond [11,12], where a hydrogen radical ($H^{•}$) is abstracted from the lipid chain [13]. The lipid radical can react with other molecules, giving rise to lipid hydroperoxides, which are extremely unstable and decompose themselves in secondary products [12,13]. The initiation phase requires an initiator molecule, such as iron or copper ions [8], reactive nitrogen species (RNS), or reactive oxygen species (ROS) [13].

The increased presence of reactive oxygen species entails the pathological process of oxidative stress, which has been related to several diseases [14] such as cancer, cardiovascular and neurodegenerative diseases [15]. As the brain is an organ with a huge energy consumption, it is more susceptible to oxidative stress. In addition, it is rich in phospholipids, and is especially enriched in PUFAs [16]. Oxidative stress generation can originate from different sources, such as NADPH oxidase enzymes [17,18], monoamine oxidase [19], peroxisomes, and lastly, the mitochondrial electron transport chain (mETC) [18,20,21] as the principal generator, primarily by complexes I and III [22]. 

Paraquat is a highly toxic herbicide that can lead to severe brain damage [23] due to its interference with NADP+ reduction [24,25]. It has also been reported as a complex I and III inhibitor [24]. In the first mechanism, Paraquat ions (${\mathrm{PQ}}^{2+}$) are reduced to their mono-cation radical form, which can react with oxygen and generate ROS [24,26,27], whereas in the last case, the inhibition of mitochondrial complexes provokes mETC dysfunction. Thus, as the principal toxic effect of paraquat is related to oxidative stress, its effects may be partially reversed using antioxidant compounds [25,28]. Concretely α-tocopherol, the vitamin E predominant form, as well as other tocopherols and tocotrienols, have lipoperoxyl radical scavenging activity [28,29,30,31]. Vitamin E is an effective antioxidant against oxidation mediated by free radicals [28,30]. The mechanism of action of this compound consists in donating a hydrogen atom to lipoperoxyl radicals generating its own radical [32]. 

As this antioxidant compound can have beneficial effects over oxidative stress conditions, levels of oxidation subproducts [33], such as monocyte superoxide anion concentrations, have been analyzed in humans [34]. However, the effects over lipids and lipoperoxyls have not yet been examined. In this work, we test the effects of stressing human astrocytic cells with Paraquat and the modification of these changes by the antioxidant α-tocopherol. Cell viability and lipid-profiling are performed. For the latter, we combine our Cell Membrane Microarray (CMMA) technology with MALDI mass spectrometry, as it allows for the simultaneous lipid-profiling of many samples [35]. Our analysis demonstrates the importance of the lipid composition and lipid homeostasis in cell viability upon oxidative stress and opens the door to high-throughput analysis of antioxidant compounds’ effects on relevant membrane samples.

## 2. Materials and Methods

### 2.1. Drugs and Reagents

(±)-α-Tocopherol, Methyl Viologen dichloride hydrate, DMEM 1 g/L glucose, Penicillin–Streptomycin Solution Hybri-Max (P/S), Fetal Bovine Serum (FBS), L-glutamine (L-Glut), 2-mercaptobenzothiazole, and 1,5-diaminonaphthalene, trypan-blue were purchased from Sigma-Aldrich (Saint Louis, MI, USA). 

### 2.2. Cell Culture and Treatments

The human astrocytic cell line 1321N1 was seeded at 10,000 cells/cm^2^ and cultivated in 12 well plates in complete medium (DMEM medium with 1 g/L glucose, 5% Fetal Bovine Serum (FBS), 1% L-glutamine (L-Glut) and 1% Penicillin-Streptomycin Solution Hybri-Max (P/S)) for 24 h at 37 °C, 5% CO_2_ and constant humidity. The cells were habituated to low serum conditions (DMEM 1 g/L glucose, 0.2% Charcoal-treated FBS, 1% L-Glut, and 1% P/S) for 12 h before treatments began. After adaptation, a pre-treatment of 3 h with or without α-Tocopherol (1 µM), prepared in a low serum medium, was performed before treatment, with or without Paraquat (500 µM), in low serum conditions for different times (24, 48, 72, 96, and 120 h). Cell viability assays were performed after each time interval as described below. 

### 2.3. Viability Assay

To analyze the cellular viability in every condition and time point, Trypan-blue viability assays were performed. The cells were detached from the culture well by a mechanical method to avoid enzymatic treatments (changing media for PBS at 4 °C and tapping the plate laterally). The cell suspension was transferred to a microtube and diluted 1:1 with 0.4% Trypan blue solution. The cells were counted using a Neubauer chamber in an inverted microscope Olympus CKX41 (Olympus Corporation, Tokyo, Japan). Dead cells (with a compromised membrane) were stained in dark blue. The percentage of live cells, with respect to total cells, was calculated in order to determine the best time for MS analysis.

Viability data handling and analysis was carried out using Excel and GraphPad software (version 9.2). Briefly, cell viability data were presented as a percentage of cell growth. The identification of outliers was carried out by applying the following formulas:$$Y_{1}=\bar{X}-\mathrm{DF}*\mathrm{SD}\;\;\;\;\;Y_{2}=\bar{X}+\mathrm{DF}*\mathrm{SD}$$

* SD = standard deviation; DF = deviation factor; 

Points were identified as outliers and excluded if Y_1_ was higher than the point analyzed or Y_2_ was lower than the point examined. We used a deviation factor of 1.25 in our analysis. The data were expressed as means of independent data points ± S.E.M. The results were analysed using one-way two-tailed two-way ANOVA with Tukey’s post hoc. Statistical differences were indicated with *p*-values ≤ 0.05.

### 2.4. Cell Membrane Extraction and CMMA Fabrication

In order to fabricate Cell Membrane Microarrays (CMMA), the 1321N1 cell line was seeded at 15,000 cells/cm^2^ and cultivated in 25 cm^2^ culture flask following the same conditions explained previously. The cells were cultivated until 80% confluency was reached (obtaining 10^6^ cells per flask), and then treated as described previously. The cells were detached from the flasks and homogenized using a Teflon-glass grinder (Heidolph RZR 2020, Schwabach, Germany) in 20 volumes of homogenization buffer (1 mM EGTA, 3 mM MgCl_2_, and 50 mM Tris-HCl, pH 7.4). The crude homogenate was subjected to a 1500-rpm centrifugation (AllegraTM X 22R centrifuge, Beckman Coulter, CA, USA) for 5 min at 4 °C, and the resulting supernatant was collected and centrifuged at 18,000× *g* (Microfuge^®^ 22R centrifuge, Beckman Coulter, CA, USA) for 15 min (4 °C). With this protocol, a fraction enriched in plasma membrane and membranes from internal organelles, including mitochondria, was obtained, as it has been demonstrated by detection of GLUT-4 transporter and Insulin Receptor β subunit [36,37], and in previous studies by cytochrome C detection [36], mETC enzymes and acetylcholinesterase activity [18,38,39]. The tubes were finally decanted, and the pellets were frozen at −80 °C, with the exception of one aliquot, which was used to determine the protein concentration. The protein concentration was determined by the Bradford method and adjusted to a final concentration of 5 mg/mL [40,41]. 

The membrane homogenates were resuspended in buffer and printed onto glass slides using a non-contact microarrayer (Nanoplotter NP 2.1, Gesim Bioinstruments, Radeberg, Germany) using a solenoid tip, placing 2 replicates of each sample (30 nL/spot), into pre-activated glass microscope slides. Printing was carried out under controlled humidity (relative humidity 60%) at a controlled temperature of 4 °C. The CMMAs were stored at −20 °C until usage. The CMMAs were validated before usage by different methods, including Bradford staining, for protein determination, enzyme activity assays (NADH oxidoreductase, succinate dehydrogenase, and cytochrome c oxidase), and radioligand binding assays [18,38,39,40].

### 2.5. MALDI-MS Lipidomic Analysis

The Cell Membrane Microarrays were covered with a suitable matrix with the aid of a standard glass sublimator (Ace Glass 8233,NJ, USA), producing a uniform film of approximately 0.2 mg/cm^2^ [35]. For positive-ion and negative-ion modes, 2-mercaptobenzothiazole (MBT) and 1,5-diaminoaphtalene (DAN) were used, respectively [35,42]. The CMMAs were scanned, as in the MALDI imaging experiment. The area of the array was explored following a grid coordinate separated by 250 μm; as each spot has a diameter of 280 μm, six pixels were recorded at each spot. The mass spectrometer used was an LTQ-Orbitrap XL (Thermo Fisher Scientific, Waltham, Massachusetts, USA), equipped with a MALDI source with a N_2_ laser (60 Hz, 100 µJ/pulse maximum power output). The laser spot is an ellipsoid of approximately 50–60 μm × 140–160 μm. Two microscans of 10 shots/pixel were used, with a laser power output of 20 µJ for MS+ and 30 µJ for MS- and a resolution of 250 μm. Data loading included spectra normalization by total ion current (TIC), spectra alignment, and peak picking, filtering all the *m/z* with intensity < 0.5% of the strongest peak in the spectrum. 

The MALDI spectra were aligned using MATLAB (Matworks, Natick, Massachusetts, USA), lipids assignment was performed using the Lipid Maps LMSD database. For the MALDI data analysis, the MS+ and MS- data were normalized separately and then analyzed together. The matrix peaks and isotopic distribution were removed, and the remaining peaks were normalized against their total ion current (TIC). The MS+ and MS- data were standardized using the z-score method and using the following formula:$$\;Z=\frac{x-\mu }{\sigma }$$

X: observed value; μ: mean of the sample; σ: standard deviation of the sample.

## 3. Results

### 3.1. Viability of Human Astrocytic Cells upon Different Treatments

The characterization of cell viability in the cultures was performed in order to decide the conditions in which to study the membrane lipid changes by CMMAs. The culture medium was maintained along the treatments, so that the cells would be entering senescence. The cultures were monitored until all of the cells were dead.

Paraquat treatment decreased cell viability at all time points, with respect to the control, changing from 83% to 68% at 24 h (Figure 1B, C). Moreover, with α-tocopherol pre-treatment, the viability improved significantly at 24 h (96%) (Figure 1C). At 72 h, the Paraquat-induced mortality reached 90% (10% viability), and the viability was rescued (up to 40%) if the cells were pre-treated with α-tocopherol. No viability differences between the control and the 3-h exposure to α tocopherol were observed when not combined with Paraquat. The cells in the control conditions reached death (98.6 %) at 96 h, while those treated with α-tocopherol only or with a pre-treatment of α-tocopherol, followed by Paraquat, survived longer (reaching death at 120 h).

Differences were observed at 24, 48, and 72 h between the Paraquat-treated and control situations and between the Paraquat-treated cells with or without α-tocopherol pre-treatment. The time chosen for mass spectrometry was 24 h, at which point viability is still not much compromised and the quantitative differences between treatments are larger. 

### 3.2. Lipidomic Analysis in Cell Membrane Microarrys Is Able to Reveal Paraquat-Triggered Changes in Human Astrocytic Membranes

We used CMMAs composed of membranes from the human astrocytic cell line 1321N1, obtained in three different conditions: Control, Paraquat (500 µM, 24 h), and Paraquat preceded by α-tocopherol (1 µM, 3 h). Membrane arrays were developed for MALDI mass spectrometry analysis of their lipid fingerprint. The differences between the lipid fingerprint were analyzed between membranes from the control and Paraquat-treated cells, and between membranes in Paraquat conditions with or without α-tocopherol pre-treatment. Both MS+ and MS- modes were performed. Specific lipid adducts were detected in the Paraquat-treated samples, in contrast to the control situation; 66 lipid adducts were only present in the Paraquat-treated membranes with respect to the control situation. Nevertheless, the vast majority of the lipid adducts present in the Paraquat condition changed when the cells were pre-treated with α-tocopherol.

The membranes from the Paraquat-treated cells displayed very long chain ceramides (C > 26) Cer 40:0; O3, Cer 40:1; O3, and Cer 40:2; O3 (red arrows in Figure 2A), which were absent in the control membranes, and sphingomyelins generally increased their presence in the Paraquat treated samples (Figure 2A). 

Long-chain glycerophosphates (PA with C > 26) (Figure 2B) also displayed a general increase, particularly evident in PA 40:6, with a very long chain with more than three unsaturations. Lipids of the phosphoethanolamines family are present as normal and as the ether version (PE and PE O-), where one of the fatty acids is connected to the glycerol molecule by an oxygen atom. In this regard, PE 36:3, PE 38:3, and PE O- 36:2 were only present in the Paraquat-treated membranes (red arrows in Figure 2B), while PE 36:2 appeared to decrease (−17.62% ± 7.2), as was the case for other PEs (Figure 2B). Moreover, two similar lipidic species with opposite behavior, upon Paraquat treatment, were observed in pairs differing in their degree of unsaturation: e.g., PE 38:2 showed a decrease of −35.7% ± 1.4, whereas PE 38:3 is one of the species only present in Paraquat treated samples. Finally, oxidated PE-O species of more than 30 carbons generally increased with Paraquat treatment. Very long chain phosphocholines (Figure 2C) are increased due to Paraquat treatment: the more unsaturation, the higher the change rates. In addition, the lysophosphocholines (lipid adducts with only one fatty acid) LPC 16:0 and LPC 18:1, presented an increase of 138% ± 25 and 261.8% ± 43.2, respectively, whereas the LPC 16:0 ether form appeared only upon Paraquat treatment (red arrows in Figure 2C). Phosphocholines with one or two unsaturations were increased in the membranes from the Paraquat treated cells, while their saturated forms were reduced or absent: e.g., PC32:1 (189.29% ± 28.1) and PC 32:2 (207.12% ± 27.0) compared with PC 32:0 (−35.74% ± 1.4). Finally, PI adducts were present as very long chain lipids, with a general increase observed in the forms with more unsaturations, while their saturated form or those with lower unsaturations were reduced or absent. (Figure 2C). Thus, the lipidomic analysis revealed a distinguishable fingerprint in cell membranes due to paraquat treatment, that we can identify using our CMMAs.

### 3.3. Lipidomic Analysis in Cell Membrane Microarrys Is Able to Reveal the Effect of α-Tocopherol Pre-Treatment on the Paraquat-Triggered Signature

The membranes from the Paraquat-treated cells with α-tocopherol pre-treatment, showed that unsaturated, very long ceramides Cer 40:1;O3 (−8.77% ± 4.9) and Cer 40:2;O3 (−35.37% ± 23.94) seemed to decrease, while their saturated version Cer40:0; O3 presented an increase of 271%. In sphingomyelins, SM 35:1;O2, SM 36:2;O2 and SM 41:2;O2 appeared only in the α-tocopherol pre-treated cells (red arrow in Figure 3A). In addition, a general increase was observed in the sphingomyelin adducts with one or two unsaturated ions (Figure 3A), while saturated ones remained absent. 

In contrast, saturated glycerophosphates generally increased; the lower the number of double bonds, the higher the change rate (Figure 3B), while phosphoglycerols showed a general decrease due to the antioxidant pre-treatment. Examples of two similar lipidic species with opposite behavior upon α-tocopherol pre-treatment also occurred. Such is the case of PG 36:1 (15.49% ± 20.32), which appeared to increase, whereas PG 36:2 did not change (−3.81% ± 3.84). This pattern can also be observed between the different phosphoethanolamine adducts; ether PE generally decreased, while normal adducts displayed a mild increase. The exception is PE O-40:5, an ether PE only present when Paraquat treatment was preceded by α-tocopherol.

With the antioxidant treatment preceding Paraquat, some very long chain phosphocholines (PC 35:4, PC 35:5) disappeared from the membranes (black arrows in Figure 3C). Ether phosphocholines generally showed a slight increase, with the exception of PC O-40:5, which was only present upon α-tocopherol pre-treatment. Lysophosphocholines between 16 and 18 carbons slightly increased (Figure 3C). Finally, the PI adducts showed a general, mild decrease, whereas LPI 20:4 seemed to increase (81.3% ± 41.41) (Figure 3C).

To summarize, clear differences triggered by Paraquat in the lipid fingerprint were detected in our system in ceramides, sphingomyelins, glycolipids, and unsaturated phospholipids. In contrast, unsaturated ceramides and phospholipids were reduced when the cells were treated with α-tocopherol before the Paraquat stress (Figure 4). 

## 4. Discussion

In the present study, we describe a method based on CMMA to observe lipidome changes in membranes from brain cells that are triggered by prooxidant and antioxidant treatments. The starting point is that α-tocopherol, with antioxidant properties against membrane lipid peroxidation, could protect from Paraquat damage [43]. We found that cell viability is enhanced when an α-tocopherol treatment precedes exposure to Paraquat. It is known that tocopherol and tocotrienol compounds prevent lipid peroxidation due to their ROS scavenger properties [44], protecting unsaturated fatty acids and lipid mediators [31]. In addition, α-tocopherol regulates the expression of genes implicated in apoptosis or antioxidant defenses, such as Bcl2-L1 or γ-glutamylcystein synthetase [44,45], or in lipid homeostasis, such as phospholipase A2 [46], which has 1-O-acyl ceramide synthase activity, using ceramides as an acceptor [47]. Thus, we predicted that lipid changes due to Paraquat exposure can be mitigated by this antioxidant pre-treatment. 

We show that changes in the membrane lipid fingerprint between experimental conditions can be analyzed with our CMMA technology, using a significantly small amount of material per sample. This allows for the production of thousands of microarrays with negligible total sample amounts. Moreover, using this high-throughput technology allows for lipidomic analysis to be performed in many samples at the same time, therefore opening the path to future large-scale screenings. CMMAs have previously been used in different lipidomic studies to analyze the lipid fingerprint of nerve and peripheral tissue in animal models [35,40]. An additional advantage of this technology is that protein expression or activity assays can be performed, as the protein-lipid relation and protein functionality is maintained [35,39,40,41,42]. Thus, the combination of membrane microarray technology with mass spectrometry results in a powerful technique to determine the effects of different compounds over lipid composition. 

It is known that the ceramide content remains low in non-dividing cells, but can increase due to exposure to stress conditions, such as serum starvation, chemical compound exposure, or oxidative stress conditions [48]. These sphingolipids, composed of sphingosine group and fatty acid, are primarily produced by de novo synthesis [49], the salvage pathway, or sphingomyelinase pathways [50]. We are able to detect an increase in ceramides in Paraquat-treated membranes, as all adducts increase their presence, whereas, with α-tocopherol pre-treatment, the unsaturated ceramides decrease with respect to their values in Paraquat treatment, or remain absent. These bioactive lipid adducts can participate in diverse signaling pathways themselves or by their hydrolyzation to sphingosine through ceramidase enzymes [51]. The increased levels observed in the membranes from the Paraquat-treated cells, with respect to the control, can act as second messengers in apoptosis pathways [52,53], increasing cell death, as it is observed in Figure 1. Moreover, the increase in ceramide production might be due to an activation of the de novo pathway, and this activation may lead to an increase in other sphingolipids by ceramides transformation, such as sphingomyelins [49] by the action of the SMS enzyme. However, the detected ceramides had ultra-long chain (ULC) fatty acids, the presence of which is related to inflammation and different diseases [54], including neurodegenerative disorders [54,55]. With α-tocopherol exposure, sphingomyelin adducts, such as SM 35:1; O2, either increase or appear as new species, with respect to the membranes from cells treated only with Paraquat. In contrast, some ceramide species decrease with antioxidant pretreatment (Figure 3), which can lead to higher cell survival (Figure 1). 

Phospholipids are particularly sensitive to oxidative stress exposure. Phosphoethanolamines and their ethers present changes upon Paraquat treatment. The observed increase in phosphoethanolamine ethers (PE O-) might be an adaptive response of the cell as they have antioxidant activity [56]. They can be precursors of plasmalogen, a lipid species whose oxidative products do not propagate lipid peroxidation [57], protecting other phospholipids, lipids, and lipoproteins from oxidative stress [58]. The observed decrease in most ether PE species when α-tocopherol precedes Paraquat treatment can reflect a return to basal conditions, where this antioxidant mechanism is not necessary [59]. The antioxidant properties of the PE O- species are mainly attributed to the preferential oxidation of the vinyl ether bond, which results in the protection of the polyunsaturated fatty acids [60]. In addition, in agreement with these observations, Paraquat treatment results in greater increases in ether forms of phosphatidylcholines, such as LPC O-16:0 and PC O-32:2, than their non-ether analogs (Figure 2C). Furthermore, as shown in other studies, [59], an increased presence of PE 38:3 is observed in cell membranes upon Paraquat treatment. These species only appear in Paraquat-treated samples with or without pre-treatment with α-tocopherol. Nevertheless, these lipid adducts containing long-chain polyunsaturated fatty acids (LCPUFAs) intervene in brain inflammatory reactions. In Paraquat-treated membranes, PCs containing LCPUFAs are increased, while with α tocopherol pre-treatment, some of them disappear [61].

We have detected an increase in LPI 20:4 when Paraquat treatment is preceded by an α-tocopherol pre-treatment. This lipidic adduct, which contains arachidonic acid (AA), might increase in the membranes due to lower AA release from the membrane phospholipids. Such an effect can be mediated by a reduction in phospholipase A2 activity [62], as α-tocopherol is known to be a modulator of phospholipase A2 [46] or by an increase in phospholipase A1 or 2-acyl LPI transferase activity [63]. In addition, excessive free AA has been related to neurotoxic effects [64]; thus, the higher relative content of lipid adducts with AA may be related to a lower free AA content. This lipid adduct can be an agonist of GPR55, a G-protein coupled receptor, whose biological activities include the modulation of immune cells and insulin secretion, and also have a potential mitogen activity in cancer cells [63]. In contrast, lysophosphocholines stimulate intracellular ROS production and ATM/Chk2, ATR/Chk1, as well as Akt activation in endothelial cells [65]. LPC 16:0, which contains palmitoyl molecule, is present in the samples treated with Paraquat and has a slight increase when samples are pre-treated with α-tocopherol. As palmitoylation is particularly important for cell membrane stabilization [66,67], its increase in our samples may support cellular viability (Figure 1).

In conclusion, the data obtained from the MALDI mass spectrometry performed in our CMMA system on astrocytic human cell membranes after prooxidant and antioxidant treatments provide distinguishable and meaningful lipid fingerprints. These differences emerge from the relative increase in ultra-long chain glycerophospholipids, unsaturated ceramides, and lysophospholipids, caused by Paraquat treatment. Tocopherol pre-treatment changes these effects by reducing the presence of unsaturated ceramides, PIs, and PE adducts, and increasing LPC and LPI, which contain palmitoyl and arachidonic acid, respectively. A correct membrane lipidome is essential for proper membrane fluidity and functionality; therefore, the effects of antioxidant compounds over them might exert an important influence on cell viability. Moreover, CMMA technology allows for the performance of this MS analysis over membranes with a negligible sample amount, in contrast to usual MS. Future high-throughput studies will allow the simultaneous identification of compounds influencing lipid profiles and their adverse or beneficiary effects on cell membranes. 

## Figures and Tables

**Figure 1 antioxidants-11-02440-f001:**
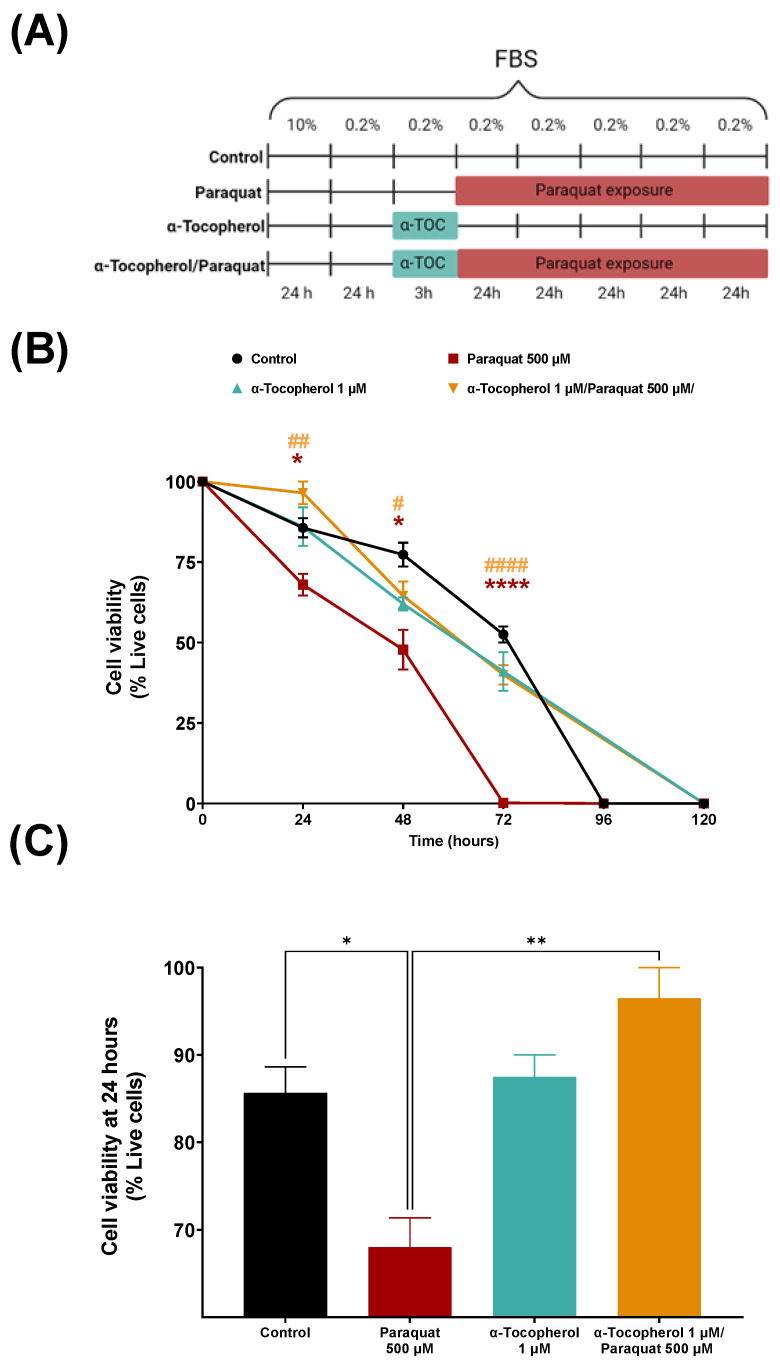
Effects of Paraquat on 1321N1 cells viability with or without pre-treatment with α-tocopherol. (**A**) Time course and design of treatments for the evaluation of cell culture viability. (**B**) Time course of cell viability in control conditions and after treatment with Paraquat (500 µM) with or without a pre-treatment with α-tocopherol (1 µM). Measurements were performed every 24 h using trypan blue cell viability assay. Asterisks symbols (*) refer to post-hoc comparisons with respect to control; hash symbols (#) refer to comparisons with respect to Paraquat. Colour of symbols allude to the condition been compared. (**C**) Cell viability results at 24 h expressed as percentage of live cells. Shapiro-Wilk test was performed to check the normality of distributions. One-way ANOVA analysis (two-tailed) and Tukey post-hoc were conducted, α set to 0.05. *p*-value < 0.05 (*), *p*-value < 0.01 (**), *p*-value < 0.0001 (****), moreover, *p*-value <0.05 (#), *p*-value < 0.01 (##), *p*-value < 0.0001 (####).

**Figure 2 antioxidants-11-02440-f002:**
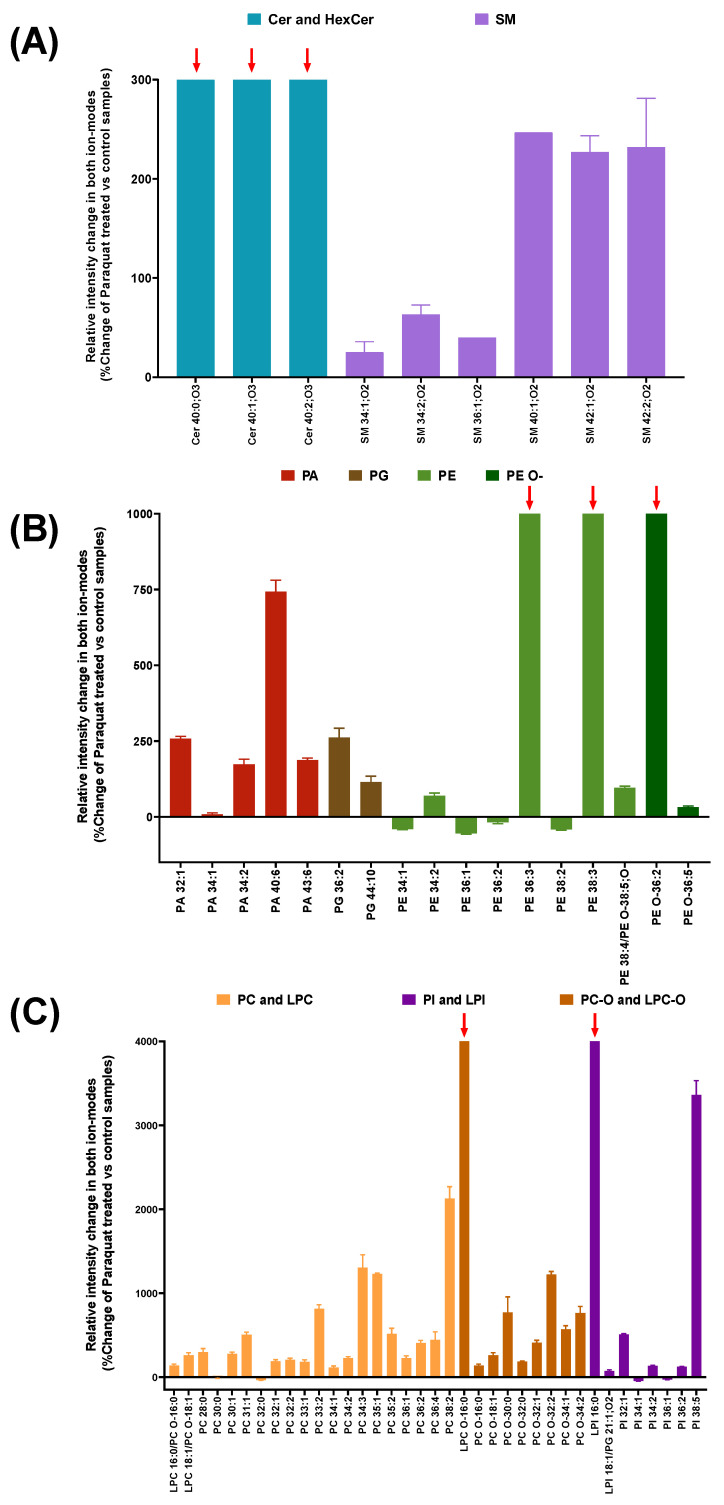
Relative intensity change in different lipid adducts when comparing Paraquat treated membranes with control samples. Data are shown as percentage of control values. (**A**) Changes in sphingolipids (Cer, HexCer and SM). (**B**,**C**)) Changes in glycerophospholipids. PA, PG, PE, and PE O- shown in (**B**). LPC, PC, and PI shown in (**C**). Abbreviations: Ceramides (Cer), hexosylceramides (HexCer), sphingomyelins (SM), glycerophosphates (PA), glycerophosphoethanolamines (PE and its ether form PE O-), glycerophosphoglycerols (PG), glycerophosphocholines and lisophosphocholines (PC and LPC), glycerophosphoinositols and lisofosfoinositols (PI and LPI), and ether forms of glycerophosphocholines and lisophosphocholines (PC O- and LPC O-). Oxygen number inside the lipid head and fatty acids in Ceramides and Sphingomyelins are indicated by O with a suffix. Bars with red arrows are only present in Paraquat-treated samples.

**Figure 3 antioxidants-11-02440-f003:**
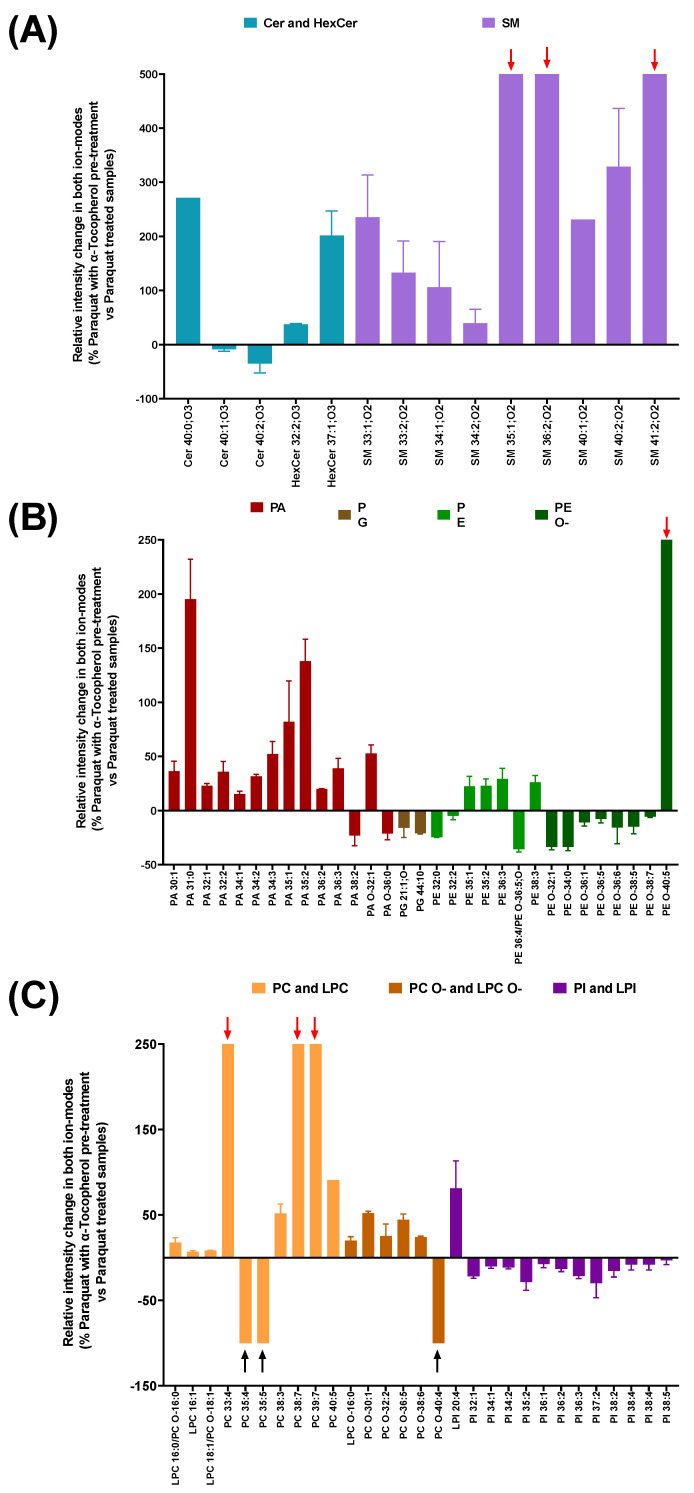
Relative intensity change in lipid adducts when comparing Paraquat-treated membranes preceded by an α-tocopherol pre-treatment with samples treated with Paraquat only. Data are shown as percentage of control values. (**A**) Change in sphingolipids (Cer, HexCer and SM). (**B**,**C**)) Changes in glycerophospholipids. PA, PG, PE, and PE O shown in (**B**). PC, PC O-, LPC, LPC O-, PI, and LPI shown in (**C**). Abbreviations: Ceramides (Cer), hexosylceramides (HexCer), spingomyelins (SM), glycerophosphatidic acid (PA), glycerophosphoethanolamine (PE), glycerophosphoglycerols (PG), glycerophosphocholines and lysophosphocholines (PC and LPC), glycerophosphoinositols and lysophosphoinositols (PI and LPI). Ether forms indicated by the O-suffix. Oxygen number inside lipid head and fatty acids in Ceramides and Sphingomyelins is indicated by O with a suffix.

**Figure 4 antioxidants-11-02440-f004:**
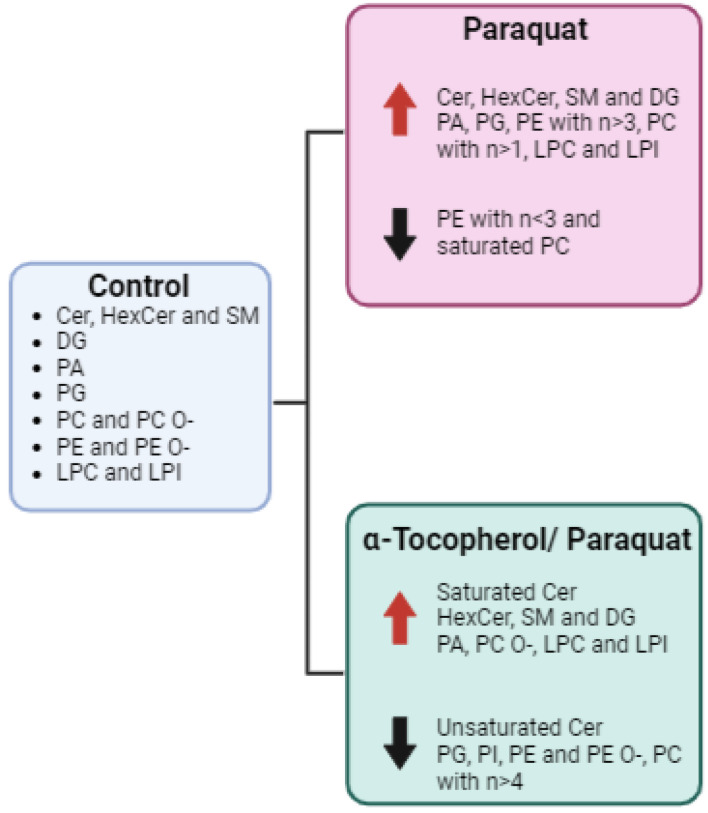
Graphic summary of lipid changes observed in CMMAs upon oxidative stress preceded or not by antioxidant treatment.

## Data Availability

The data are contained within the article.

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
