# Peer review of "Protective Actions of α-Tocopherol on Cell Membrane Lipids of Paraquat-Stressed Human Astrocytes Using Microarray Technology, MALDI-MS and Lipidomic Analysis"

_antioxidants, 2022, doi:10.3390/antiox11122440_

Round 1

Reviewer 1 Report

This manuscript is concerned with examining the antioxidant and protective effect of Vitamin E in human astrocytes stressed with paraquat. The subject area of this MS is a clear relevance and appropriate to the Journal. However a large number of concerns were evident in the paper, in particular: the lack of soundness of the Title, Abstract, Introduction and Discussion sections.

The protective effect of Vitamin E on cells submitted to oxidants is not an original finding. However, if the results from the MALDI-MS lipidomic analysis are novel, both from a qualitative (this point must be clarified) and quantitative point of view, the entire manuscript (including the title) must be improved to better show the originality and novelty of the approach.

The present version of this MS does not reach any of the standards required for publication in Antioxidants

Author Response

This manuscript is concerned with examining the antioxidant and protective effect of Vitamin E in human astrocytes stressed with paraquat. The subject area of this MS is a clear relevance and appropriate to the Journal. However, a large number of concerns were evident in the paper, in particular: the lack of soundness of the Title, Abstract, Introduction and Discussion sections.

The protective effect of Vitamin E on cells submitted to oxidants is not an original finding. However, if the results from the MALDI-MS lipidomic analysis are novel, both from a qualitative (this point must be clarified) and quantitative point of view, the entire manuscript (including the title) must be improved to better show the originality and novelty of the approach.

The present version of this MS does not reach any of the standards required for publication in Antioxidants

We would like to thank the reviewer for the constructive comments.

We have changed the title and abstract from the original draft, and also changed the approach to the article subject to clarify its originality and novelty. Changes are highlighted in the new version of the article.

Reviewer 2 Report

The manuscript investigates the effect of a-tocopherol administration on cytotoxicity and lipid profile after exposure to Paraquat of a human astrocyte cell line. The study has been performed using the Cell Membrane Microarrays (CMMA) and MALDI-TOF techniques.

The combination of these methods is interesting, but the experimental design shows some critical points.

The authors have to provide information about:

1) the number of cells used in all the experiments (cell viability and membrane extraction);

2) why they used mechanical method to detach them. In this way the Trypan-blue viability test is performed on both attached and detached cells at the same time. It would be more correct to carried out the viability analysis on detached cells and culture medium separately;

3) the membranes used for the preparation of CMMA have been obtained by centrifuging cell the supernatant obtained from crude homogenate (first centrifugation 1,550 rpm for 5 min) at 18,000 g for 15 min. The authors have to specify what types of cell membranes are present in the pellet, since after that centrifugation membranes of heavy mitochondria and microsome are not present in the pellet.

This is an important point since it influences the accuracy of the lipidomic results and the conclusions.

4) Data in Figure 1 evidence a high increase in cell mortality during the time. The authors have to explain this event. Is it due to the low serum amount in the culture medium?

Moreover, why no information about statistical analysis is present in panel B.

In Panel C, the Y axis is labelled as "Cell Toxicity", but in the legend is reported: "Cell viability expressed as percentage of dead cells". This statement has to be checked.

5) The comment to Figure 1 also includes data at 96 and 120 h, but no data at these experimental time are present in the Figure.

The authors must to revise the statement "Viability differences between Para-178 quat-treated cells, those with α-tocopherol pre-treatment, and control cells were observed 179 only at 24 hours…".

6) The Conclusions mainly focus on the positive results obtained by the combination of CMMA and MALDI-TOF techniques, and not on the possible biological impact of the lipidomic observations. This is in contrast with the title of the manuscript referring to the antitoxic effect of a-tocopherol.

Therefore, the authors have to modify their conclusions.

Author Response

REVIEWER 2

We are extremely grateful for the positive comments, we have clarified some points. Your comments and advices has made us to check an improve the article.

The manuscript investigates the effect of a-tocopherol administration on cytotoxicity and lipid profile after exposure to Paraquat of a human astrocyte cell line. The study has been performed using the Cell Membrane Microarrays (CMMA) and MALDI-TOF techniques.

The combination of these methods is interesting, but the experimental design shows some critical points.

The authors have to provide information about:

  • the number of cells used in all the experiments (cell viability and membrane extraction);

Number of cells used in viability and membrane extraction have been clarified in the paper, in lines 94-95 and 124-127 respectively.

  • why they used mechanical method to detach them. In this way the Trypan-blue viability test is performed on both attached and detached cells at the same time. It would be more correct to carried out the viability analysis on detached cells and culture medium separately;

Cells were detached using a mechanical method in order to avoid enzymatic treatments. Viability test was performed to the whole cell suspension (lines 106-108).

  • the membranes used for the preparation of CMMA have been obtained by centrifuging cell the supernatant obtained from crude homogenate (first centrifugation 1,550 rpm for 5 min) at 18,000 g for 15 min. The authors have to specify what types of cell membranes are present in the pellet, since after that centrifugation membranes of heavy mitochondria and microsome are not present in the pellet.

For membrane extraction, the protocol described by Nishiumi and Ashida (2007) and Yamamoto et.al (2016) was performed with minor alterations. With this methodology authors demonstrate the presence of cytochrome c, IR-β, GLUT-4, and β.actin indicating an enrichment of membranes from mitochondria and plasma membrane respectively. Moreover, in previous articles with the same protocol for membrane preparation, the activity of different enzymes from mitochondrial electron transport chain has been quantified (Elexpe et al 2021, Elexpe et al 2022) and other enzymatic activities, such as Acetilcholinesterase (Rienda et al 2021). (lines 134 – 137)

This is an important point since it influences the accuracy of the lipidomic results and the conclusions.

  • Data in Figure 1 evidence a high increase in cell mortality during the time. The authors have to explain this event. Is it due to the low serum amount in the culture medium?

Increased in cell mortality is due to the maintenance of cell culture medium so cells could enter in senescence (lines 175-177)

Moreover, why no information about statistical analysis is present in panel B.

Statistical information has been clarified in figure 1 panel B, and its legend.

In Panel C, the Y axis is labelled as "Cell Toxicity", but in the legend is reported: "Cell viability expressed as percentage of dead cells". This statement has to be checked.

Data in figure 1 has been changed and represented as percent of live cells over total. Text has been changed to adequate to new graphical representation (lines 189-201).

  • The comment to Figure 1 also includes data at 96 and 120 h, but no data at these experimental time are present in the Figure.

Data has been added to Figure 1 and the text modified accordingly.  

The authors must to revise the statement "Viability differences between Para-178 quat-treated cells, those with α-tocopherol pre-treatment, and control cells were observed 179 only at 24 hours…".

Larger differences in viability assay between treatments has been observed at 24 hours. Nevertheless, 24 h, 48 h and 72 h presented differences both between Paraquat treated cells and control ones and between Paraquat treatment with or without the α-tocopherol pre-treatment (lines 197-200).

6) The Conclusions mainly focus on the positive results obtained by the combination of CMMA and MALDI-TOF techniques, and not on the possible biological impact of the lipidomic observations. This is in contrast with the title of the manuscript referring to the antitoxic effect of a-tocopherol.

Therefore, the authors have to modify their conclusions.

Connections between the lipid changes obtained and expected biological outcomes have been added to the discussion (highlighted) and conclusions have been modified in lines 376-386.

Round 2

Reviewer 1 Report

The Ms has been improved by the authors to better evidence the soundness of the research. The Ms is now suitable for  publication in Antioxidants

Author Response

Reviewer 1

The Ms has been improved by the authors to better evidence the soundness of the research. The Ms is now suitable for  publication in Antioxidants

English language and spelling has been check and improve, changes are highlighted in yellow.

Reviewer 2 Report

The authors revised the manuscript according to the suggestions, this improving it.

A minor correction has to be made: the information about cell seeding has to be modified from 10.000 or 15.000 to10,000 and 15,000.

After this modification, the manuscipt is suitable for publication.

Yours sincerely

giuliana muzio

Author Response

Reviewer 2

The authors revised the manuscript according to the suggestions, this improving it.

A minor correction has to be made: the information about cell seeding has to be modified from 10.000 or 15.000 to10,000 and 15,000.

After this modification, the manuscipt is suitable for publication.

Yours sincerely

giuliana muzio

Numbers have been changed to 10,000 and 15,000. Also, English language and spelling has been checked in order to improve it. Changes are highlighted in yellow.
